# Prevention of L-Dopa-Induced Dyskinesias by MPEP Blockade of Metabotropic Glutamate Receptor 5 Is Associated with Reduced Inflammation in the Brain of Parkinsonian Monkeys

**DOI:** 10.3390/cells11040691

**Published:** 2022-02-16

**Authors:** Marc Morissette, Mélanie Bourque, Marie-Ève Tremblay, Thérèse Di Paolo

**Affiliations:** 1Centre de Recherche du CHU de Québec, Axe Neurosciences, 2705, Boulevard Laurier, Quebec, QC G1V 4G2, Canada; marc.morissette@crchudequebec.ulaval.ca (M.M.); melanie.bourque@crchudequebec.ulaval.ca (M.B.); 2Division of Medical Sciences, University of Victoria, Victoria, BC V8P 5C2, Canada; evetremblay@uvic.ca; 3Department of Biochemistry and Molecular Biology, The University of British Columbia, Vancouver, BC V6I 1Z3, Canada; 4Faculté de Pharmacie, Université Laval, Pavillon Ferdinand-Vandry, 1050, Avenue de la Médecine, Quebec, QC G1V 0A6, Canada

**Keywords:** inflammation, L-Dopa-induced dyskinesias, Parkinson, MPEP, L-Dopa, MPTP monkey, GFAP, Iba1, CD68, basal ganglia

## Abstract

Proinflammatory markers were found in brains of Parkinson’s disease (PD) patients. After years of L-Dopa symptomatic treatment, most PD patients develop dyskinesias. The relationship between inflammation and L-Dopa-induced dyskinesias (LID) is still unclear. We previously reported that MPEP (a metabotropic glutamate receptor 5 antagonist) reduced the development of LID in de novo MPTP-lesioned monkeys. We thus investigated if MPEP reduced the brain inflammatory response in these MPTP-lesioned monkeys and the relationship to LID. The panmacrophage/microglia marker Iba1, the phagocytosis-related receptor CD68, and the astroglial protein GFAP were measured by Western blots. The L-Dopa-treated dyskinetic MPTP monkeys had increased Iba1 content in the putamen, substantia nigra, and globus pallidus, which was prevented by MPEP cotreatment; similar findings were observed for CD68 contents in the putamen and globus pallidus. There was a strong positive correlation between dyskinesia scores and microglial markers in these regions. GFAP contents were elevated in MPTP + L-Dopa-treated monkeys among these brain regions and prevented by MPEP in the putamen and subthalamic nucleus. In conclusion, these results showed increased inflammatory markers in the basal ganglia associated with LID and revealed that MPEP inhibition of glutamate activity reduced LID and levels of inflammatory markers.

## 1. Introduction

Parkinson’s disease (PD) is the second most common age-related neurodegenerative disorder after Alzheimer’s disease, affecting approximately 1–3% of the population over age 65 [1]. PD is characterized clinically by resting tremor, bradykinesia, rigidity, gait disturbances, and pathologically by the selective and progressive degeneration of midbrain dopamine (DA) neurons in the substantia nigra *pars compacta* (SNc) and their projections to the striatum, leading to a significant decrease in DA content in the striatum [2]. Although approximately 5–10% of PD cases can be directly attributable to genetic factors associated with mutations of some genes [3], the aetiology of idiopathic PD so far remains unknown. Until now, there is no cure for PD, and treatments remain symptomatic. The clinical use of L-3,4-dihydroxyphenylalanine (L-Dopa), the DA precursor, combined with an L-Dopa decarboxylase inhibitor to prevent L-Dopa degradation in the periphery, was introduced more than 50 years ago and still remains the gold standard treatment [4]. However, after long-term use, its beneficial effects are hampered by motor fluctuations and L-Dopa-induced dyskinesias (LID) that are clinically difficult to manage, limiting the quality of life in PD patients [5]. The pathophysiology of LID is complex, involving pathological receptor and downstream molecular alterations affecting presynaptic and postsynaptic signal transduction in nigrostriatal DA neurotransmission, as well as other nondopaminergic neurotransmitter systems [5]. An overactive glutamatergic neurotransmission in the basal ganglia is well documented in animal models of PD, and the emergence of LID are classically associated to a maladaptive plasticity of corticostriatal synapses in response to L-Dopa treatment [5,6,7].

To counteract overactive and aberrant glutamate receptor signaling in striatal medium spiny neurons and the associated dyskinesias [5], the use of ionotropic glutamatergic receptor antagonists (iGlu), in particular, N-methyl-D-aspartate (NMDA) receptors, was considered as an antiglutamatergic treatment [8]. However, despite a positive therapeutic potential in several preclinical studies, many harmful side effects were observed, hampering the clinical development of these substances [9]. Aside from some short-term benefit of amantadine, a low-affinity noncompetitive NMDA receptor antagonist and, to a lesser extent, clozapine, there is no other pharmacological alternative available to treat this adverse effect [10,11]. However, in recent years attention was paid to metabotropic glutamate (mGlu) receptors and especially the mGlu5 receptor (group I) to reduce this overall excitatory drive in the basal ganglia. The mGlu5 receptor is positively coupled to phospholipase C (PLC) and the inositol 1,4,5-trisphosphate/calcium/protein kinase C (IP3/Ca^2+^/PKC) pathway. It is widely expressed within the basal ganglia and its postsynaptic localization on striatal medium spiny neurons of the direct and indirect pathways makes it an interesting target for modulating activity at the level of corticostriatal synapses [12]. Interestingly, a close interaction of mGlu5 receptor with NMDA and DA D1 receptors signaling pathways, two components of the basal ganglia suspected of being involved in the onset of LID, was demonstrated in the basal ganglia [13,14,15,16]. Accordingly, upregulation of mGlu5 receptor levels were observed in the striatum of dyskinetic 1-methyl-4-phenyl-1,2,3,6-tetrahydropyridine (MPTP)-lesioned monkeys and in parkinsonian patients with motor complications [17,18,19,20].

In addition, a large number of studies in 6-hydroxydopamine (6-OHDA)-lesioned rats and MPTP monkeys showed that coadministration of negative allosteric modulators (NAMs) of mGlu5 receptors such as 2-methyl-6-(phenylethynyl) pyridine (MPEP), 3-[(2-methyl-1,3-thiazol-4-yl) ethynyl] pyridine (MTEP), Fenobam, AFQ056/Mavoglurant), ADX48621/Dipraglurant), or MRZ-8676 with L-Dopa maintained antiparkinsonian effect while inhibiting or reducing the duration and intensity of LID (see review: [21]). Interestingly, the antidyskinetic effect of MPEP in the same MPTP monkeys used in the present experiment was associated with a normalization of striatal DA D1 and D2 receptors, their associated signaling proteins (ERK1/2 and Akt/GSK3β) and neuropeptides (preproenkephalin, preprodynorphin), together with adenosine A_2A_ receptors, α7 nicotinic acetylcholine receptors, the mGlu5 and mGlu2/3, and the ionotropic α-amino-3-hydroxy-5-methyl-4-isoxazolepropionic acid (AMPA) and NMDA NR1/NR2B glutamate receptors [17,22,23,24,25]. Moreover, it was demonstrated that the systemic administration of MPEP or MTEP alone in rodents and nonhuman primate models of PD mediated neuroprotection of the nigrostriatal dopaminergic system [26,27,28,29]. Together, these results suggest that the neuroprotective effects of NAMs of mGlu5 receptors on dopaminergic neurons could be mediated at least in part by a modulation of the inflammatory activity of glial cells since microglia and astrocytes express mGlu5 receptors [30,31,32,33] and upregulation of these receptors was observed in reactive astrocyte in different models of brain injury [34,35]. However, studies in primary cultures of microglia and microglial cell lines indicate that stimulation of mGlu5 receptors induces anti-inflammatory effects by decreasing the neurotoxic effects of microglia [31,36,37,38,39,40,41,42] whereas inhibition of mGlu5 receptors in BV-2 microglial cell line enhances cellular stress and production of inflammatory mediators [43]. By contrast, chronic treatment with MPEP in an MPTP-induced PD rat model was reported to decrease microglial upregulation of major histocompatibility complex (MHC) II reactivity in the SNc [44].

Since the first postmortem description of reactive microglia in the SN of patients with PD, a plethora of preclinical and clinical studies notably conducted in the postmortem brain of PD patients have reinforced the hypothesis that inflammation could play a pivotal role in the pathophysiology of PD, with both microglia and astroglia involved in the neurodegenerative process of DA neurons (see reviews: [45,46]). Furthermore, several preclinical studies suggested that treatment with L-Dopa, primarily in advanced stages of PD, could aggravate inflammation in the brain via the release of proinflammatory mediators by microglia, thereby resulting in a propitious environment for the development and expression of LID [47]. In fact, an increased expression of genes implicated in the inflammatory cascade was detected in the striatum of 6-OHDA-lesioned rats with abnormal involuntary movements (AIMs) induced by L-Dopa treatment and in the striatum of MPTP-lesioned marmosets with LID [48,49]. While inflammatory responses seem to be associated with L-Dopa treatment, it is still unclear whether they are dependent on the dyskinetic outcome of the L-Dopa treatment and causally linked to LID.

In the present study, we thus investigated the inflammatory response in different nuclei of the basal ganglia (caudate nucleus, putamen, globus pallidus, substantia nigra and subthalamic nucleus) of de novo MPTP-lesioned cynomolgus female monkeys in relation to the development of LID using western blot measures of glial fibrillary acidic protein (GFAP) and ionized calcium-binding adapter molecule 1 (Iba1) as markers of astrocyte and microglia, respectively. Levels of the phagocytosis-related receptor CD68, a glycoprotein associated with the phagolysosomal activity of microglia, were also measured since increased expression of microglial CD68 was observed in SN of PD patients as well as in mouse and rat models of PD overexpressing α-synuclein. This was observed early during the pathology, either prior or coinciding with cell death [50,51,52]. Inflammatory activity was also investigated in the primary motor cortex (M1) and in dorsolateral prefrontal cortex (PFCd) to determine if the inflammatory process observed in the pathophysiology of PD and LID could affect other regions outside the basal ganglia circuit not known to play a role in the neurodegeneration of DA neurons. We also investigated the nucleus accumbens to compare other dopaminergic pathways than nigrostriatal. Moreover, we evaluated the inflammatory response in monkeys with LID treated with the NAM of the mGlu5 receptor MPEP that reduced the development of dyskinesias in these monkeys [23].

## 2. Materials and Methods

### 2.1. Animals and Treatments

Drug-naive ovariectomized female cynomolgus monkeys (*macaca fascicularis*) were used for the present experiment. Handling of primates by qualified personnel was in accordance with the recommendations of the National Institute of Health Guide for the Care and Use of Laboratory Animals. All procedures implemented to minimize discomfort and provide an enriching environment for animals were reviewed and approved by the Institutional Animal Care Committee of Laval University. Continuous infusion of MPTP via subcutaneous Alzet osmotic minipumps (Durect Corporation, ALZET Osmotic Pumps, Cupertino, CA 95014-4166, USA) (0.5 mg/24 h) was used to render monkeys parkinsonian until animals develop stable parkinsonian syndrome, which was achieved after 6.6 months on average. After this period of stabilization, monkeys were treated once daily for one month with L-Dopa/benserazide 100/25 mg capsule per os alone (Prolopa, Hoffmann–La Roche, Basel, Switzerland; a mixture of 100 mg of L-Dopa and 25 mg benserazide; *n* = 5 monkeys) or with the combination of L-Dopa/benserazide (100/25 mg) and MPEP (from Novartis Pharma AG, Basel, Switzerland; 10 mg/kg, administered once daily 15 min prior to L-Dopa; *n* = 5 monkeys). Since the mean weights were similar for these two MPTP groups, the same dose of (based on body weight) 100 mg L-Dopa/25 mg benserazide once daily for one month was given to the MPTP-lesioned monkeys of both groups. Four intact monkeys that serve as normal controls as well as four MPTP-lesioned monkeys that received saline only were also included in the present experiment for subsequent biochemical analysis (Figure 1).

### 2.2. Motor Behavior Measures

The animals were first evaluated following vehicle (sterile water) administration alone. Behavioral measures after vehicle (water) and L-Dopa/benserazide or the combination of MPEP and L-Dopa/benserazide were performed every other day for 30 days. The animals were scored every 15 min for antiparkinsonian and dyskinetic responses for the whole duration of the L-Dopa response. Independent evaluators blind to treatment assignment rated behavioral measures. The detailed behavioral assessment of these monkeys showing the effect of MPEP on the antiparkinsonian response to L-Dopa and the development of dyskinesias was previously reported [23]. At the end of experiments, all 10 drug-treated MPTP monkeys, along with the four controls and four saline-treated MPTP-lesioned monkeys, were euthanized by an overdose of sodium pentobarbital. The time interval between the last drug treatment and euthanasia was 24 h. Brains were removed, immersed in isopentane for <30 s (−40 °C), and kept frozen at −80 °C until use.

### 2.3. Striatal DA and Serotonin (5-HT) Contents

Small pieces of caudate and putamen were dissected from coronal sections of frozen brains and then homogenized with polypropylene tissue grinder in 250 μL of 0.1 M HClO_4_ at 4 °C and centrifuged at 10,000× *g* for 20 min to precipitate proteins. The supernatants were kept at −80 °C in small polyethylene tubes until time of assay, while the pellets were solubilized in 100 μL of 0.1 M NaOH for determination of protein content with a Micro BCA Protein Assay kit (Thermo Scientific, Rockford, IL, USA). For the determination of the DA and 5-HT contents in the striatal tissue, the supernatants were directly injected into the chromatograph consisting of a Waters 717 plus autosampler automatic injector (Waters Corporation, Milford, MA 01757, USA), a Waters 515 pump equipped with a C-18 column (Waters Nova-Pak C18, 3 μm, 3.9 mm × 150 cm), a BAS LC-4C electrochemical detector and a glassy carbon electrode. The mobile phase delivered at a flow rate of 0.8 mL/min was composed of 0.025 M citric acid (Sigma Aldrich Canada, Oakville, ON, Canada), 1.7 mM 1-heptane-sulfonic acid (Sigma Aldrich Canada, Oakville, ON, Canada), and 10% methanol (Fisher Scientific Company, Ottawa, ON, Canada), in HPLC grade distilled water. NaOH was added to reach a final pH of 3.98. Results were expressed in nanograms of amine per milligram of protein.

### 2.4. Tissue Preparation and Western Blot

The caudate nucleus, putamen, nucleus accumbens, external (GPe), and internal (GPi) segments of globus pallidus, whole substantia nigra (SNc and SN *pars reticulata* parts, named thereafter SN), subthalamic nucleus (STN), M1 (including regions controlling foot, trunk, and hand), and the PFCd were dissected precisely on a brain section about 1 mm thick at the level of anterior and posterior commissure from the whole left hemisphere (Figure 2). The brain tissues were homogenized in radioimmunoprecipitation assay (RIPA) lysis buffer (Santa Cruz Biotechnology, Inc., Dallas, Texas 75220, USA) supplemented with protease and phosphatase inhibitors. Solubilized homogenates were centrifuged at 13,200× *g* for 15 min. Protein contents of supernatants were quantified by a Micro BCA Protein Assay kit (Thermo Scientific, Waltham, MA, USA). Proteins were separated using 10% (CD68), 12% (GFAP) or 15% (Iba1) SDS-polyacrylamide gel electrophoresis and blotted to a polyvinylidine difluoride (PVDF) membrane. Immunodetection was performed using specific primary antibodies against GFAP (New England Biolabs, Ltd., Whitby, ON, Canada; diluted 1:1000), Iba1 (Wako Chemicals, Japan; diluted 1:1000), CD68 (New England Biolabs, Ltd., Whitby, ON, Canada; diluted 1:1000) as well as βIII-tubulin (New England Biolabs, Ltd., Whitby, ON, Canada; diluted 1:8000) and using horseradish peroxidase-coupled secondary antibodies (New England Biolabs, Ltd., Whitby, ON, Canada; diluted 1:5000). The bands were visualised using a chemiluminescent detection kit (Clarity Western ECL Blotting Substrate, Bio-Rad, Mississauga, ON, Canada). Densitometric analysis using AlphaView Image Analysis Systems (FluorChem Q system supported by AlphaView software v.3.2.2; Proteinsimple, Santa Clara, CA 95051, USA) was performed after normalizing with the expression of βIII-tubulin.

### 2.5. Statistical Analysis

Results were expressed as the mean ± standard error of the mean (S.E.M.) of two to six independent experiments. Differences between groups were analyzed by one-way analysis of variance (ANOVA) using GraphPad Prism version 9 (GraphPad Software, La Jolla, CA, USA) followed by a post-hoc analysis with the Holm–Sidak’s multiple comparison test. A log transformation of the data was performed in some statistical analyzes to homogenize variance of the groups. A simple regression model was used to determine the coefficient of correlation. A *p* ≤ 0.05 was required for the results to be considered statistically significant.

## 3. Results

### 3.1. MPEP Inhibited the Development of LID in MPTP Monkeys 

A summary of animal and behavioral data of experimental monkeys as well as striatal DA and 5-HT contents is provided in Table 1. No difference in age and weight of the monkeys was observed. Monkeys of the MPTP, MPTP + L-Dopa and MPTP + L-Dopa + MPEP groups had similar basal parkinsonian scores and similar loss of DA contents in the caudate nucleus and putamen, whereas 5-HT contents remained unchanged between experimental groups. Moreover, the survival time post MPTP of these three groups was not significantly different. Hence, the present group of monkeys allowed focusing on the effect of the lesion and treatments. As previously reported, the antiparkinsonian effect of L-Dopa was maintained in these MPTP-lesioned monkeys treated with L-Dopa + MPEP compared to the L-Dopa group [23]. The MPTP + L-Dopa + MPEP group of monkeys displayed significantly less LID compared to that of MPTP monkeys that received L-Dopa alone (Table 1 and [23]).

### 3.2. MPEP Attenuated the Increase in GFAP Levels Induced by L-Dopa in the Putamen and STN of MPTP-Lesioned Monkeys 

Possible alterations of astrocyte activity were evaluated by measuring GFAP levels in the brain of these monkeys. In the caudate nucleus at the pre-commissural level significant changes were observed (F (3,14) = 5.898, *p* = 0.0081), with significant and similar increases in the L-Dopa-treated groups of monkeys (+84%) and in L-Dopa + MPEP (+86%) compared to intact animals, whereas it did not reach statistical significance in the saline-treated MPTP monkeys (Figure 3a). Significant changes were also measured in the caudate nucleus at post-commissural levels (F (3,13) = 5.409, *p* = 0.0123), where the increase in the MPTP group (+71%) reached statistical significance (Figure 3b). A similar increase was observed in MPTP (+71%) and MPTP + L-Dopa (+73%) monkeys compared to that of intact animals (Figure 3b). An increase was also observed in L-Dopa + MPEP-treated monkeys compared to intact animals but to a lesser extent (+56%) (Figure 3b).

No change in GFAP levels was observed between all groups of animals in the nucleus accumbens (F (3,14) = 0.5947, *p* = 0.6288) (Figure 3c).

In the pre- (F(3,14) = 6.828, *p* = 0.0046) and post-commissural putamen (F (3,13) = 7.004, *p* = 0.0048), GFAP levels were increased in MPTP monkeys (pre: +58%; post: +86%) with a larger (double and triple respectively) increase seen in dyskinetic MPTP + L-Dopa monkeys (pre: +117%; post: +205%); MPTP monkeys treated with L-Dopa + MPEP (pre: +71%; post: +105%) had an increase compared to intact monkeys similar to the MPTP group (Figure 3d,e). 

No change in GFAP levels was observed between all groups of animals in the GPe (F (3,14) = 1.451, *p* = 0.2703) (Figure 4a) and the GPi (F (3,14) = 2.501, *p* = 0.1019) (Figure 4b).

In the SN, GFAP levels were significantly affected in the experimental groups (F (3,14) = 9.021, *p* = 0.0014) with increases in the MPTP + L-Dopa (+24%) and MPTP + L-Dopa + MPEP (+20%) groups compared to that of intact animals (Figure 4c); this was not significant for the MPTP group. In the STN, GFAP levels were significantly increased (F (3,14) = 6.323, *p* = 0.0062) in dyskinetic MPTP + L-Dopa monkeys (+76%) compared to intact and MPTP-treated monkeys, whereas in MPTP monkeys treated with L-Dopa + MPEP, no significant change was measured (Figure 4d). 

No change in GFAP-levels was observed between all groups of animals in the M1 (F (3,14) = 0.7371, *p* = 0.5472) (Figure 5a) and the PFCd (F (3,14) = 1.988, *p* = 0.1622) (Figure 5b). No correlation was found between the mean dyskinesia scores and GFAP levels for all the brain regions investigated in this study (data not shown).

### 3.3. MPEP Attenuated the Increase in Iba1 Levels Induced by L-Dopa in the Putamen, Substantia Nigra, and Globus Pallidus of MPTP-Lesioned Monkeys

Possible alterations of microglial activity were first investigated by measuring Iba1 contents in the brain of the experimental monkeys. Iba1 levels in the caudate nucleus at pre-commissural levels were significantly changed (F (3,14) = 6.054, *p* = 0.0073) with a decrease in MPTP (−45%) and in L-Dopa + MPEP-treated (−48%) monkeys compared to intact animals (Figure 6a). A tendency to decrease was also observed for Iba1 levels (−31%) in L-Dopa-treated MPTP monkeys compared to intact animals but did not reach statistical significance (*p* = 0.09) (Figure 6a). There was a strong positive correlation (*r* = 0.577, *p* = 0.031) between the mean dyskinesia score and Iba1 levels in the caudate nucleus at pre-commissural levels (Figure 6b). 

No change in Iba1 levels was observed between all groups of animals in the caudate nucleus at the post-commissural level (F (3,14) = 1.131, *p* = 0.3701) (Figure 6c) and in the nucleus accumbens (F (3,14) = 0.3273, *p* = 0.8057) (Figure 6d).

In the pre-commissural putamen, Iba1 levels showed significant changes (F (3,14) = 9.009, *p* = 0.0014) (Figure 6e). While Iba1 levels in MPTP monkeys were similar to those observed in intact monkeys, animals treated with L-Dopa alone showed an increase in Iba1 levels (+23% vs. intact) compared to intact and MPTP-treated monkeys (Figure 6e). This increase was prevented, with the values remaining unchanged in L-Dopa + MPEP treated MPTP monkeys (Figure 6e). There was a strong positive correlation (*r* = 0.738, *p* = 0.003) between the mean dyskinesia score and Iba1 levels in the putamen (Figure 6f) at the pre-commissural level suggesting a possible link between the onset and development of dyskinesia and inflammation. No change in Iba1 levels was observed between all groups of animals in the putamen at the post-commissural level (F (3,14) = 0.616, *p* = 0.6159) (Figure 6g). 

A similar pattern of Iba1 changes was observed in the GPe (F(3,14) = 5.454, *p* = 0.0107) and GPi (F (3,14) = 4.27, *p* = 0.0245) (Figure 7a,c). Indeed, Iba1 levels were similar in MPTP and intact monkeys (Figure 7a,c) but treatment with L-Dopa alone increased Iba1 levels compared to levels in intact (GPe: +61% and GPi: +55%) monkeys (Figure 7a,c). No significant change was observed for L-Dopa + MPEP-treated monkeys in the GPe (Figure 7a) and GPi (Figure 7b). In addition, a strong positive correlation was found between the mean dyskinesia score and Iba1 levels in the GPe (*r* = 0.581, *p* = 0.029) (Figure 7b) and GPi (*r* = 0.764, *p* = 0.001) (Figure 7d). 

In the SN, Iba1 levels also showed significant changes (F (3,14) = 4.775, *p* = 0.017) (Figure 7e). Dyskinetic monkeys treated with L-Dopa had a significant increase in Iba1 contents compared to that of intact (+37%) monkeys (Figure 7e). No significant change of Iba1 levels was observed in the vehicle treated MPTP monkeys and in the L-Dopa + MPEP-treated monkeys (Figure 7e). A strong positive correlation was also observed between the mean dyskinesia score and Iba1 levels in the SN (Figure 7f). 

No change in Iba1 levels was observed between all groups of animals in the STN (F (3,14) = 0.1573, *p* = 0.9232) (Figure 7g), M1 (F (3,14) = 1.021, *p* = 0.4131) (Figure 8a) and PFCd (F (3,14) = 0.1791, *p* = 0.9088) (Figure 8b). 

### 3.4. MPEP Attenuated the Increase in CD68 Levels Induced by L-Dopa in the Putamen and Globus Pallidus of MPTP-Lesioned Monkeys

We next examined the expression of the phagocytic marker CD68. No change in CD68 levels was observed between all groups of animals in pre- (F (3,14) = 0.050, *p* = 0.985) and post-commissural caudate nucleus (F (3,14) = 1.497, *p* = 0.258) (Figure 9a,b) and the nucleus accumbens (F (3,14) = 1.251, *p* = 0.329) (Figure 9c).

The levels of CD68 showed significant changes (F (3,14) = 3.576, *p* = 0.041) in the pre-commissural putamen with an increase only observed in MPTP + L-Dopa-treated animals (+49%) compared to intact monkeys (Figure 9d). A strong positive correlation (*r* = 0.673, *p* = 0.008) was also observed between the mean dyskinesia score and CD68 levels in the putamen at the pre-commissural level (Figure 9e). No significant change of CD68 levels was observed in the post-commissural putamen (F (3,14) = 1.325, *p* = 0.305) (Figure 9f). 

In the GPi, a significant increase in CD68 levels was observed (F (3,14) = 4.480, *p* = 0.021) in the MPTP + L-Dopa-treated animals compared to that of intact ones (+57%) (Figure 10b). In the MPTP + L-Dopa + MPEP-treated monkeys, the levels of CD68 were not significantly increased (Figure 10b). No correlation was observed between the mean dyskinesia score and CD68 levels in the GPi. No change in CD68 levels was observed between all groups of animals in the GPe (F (3,14) = 0.444, *p* = 0.725) (Figure 10a).

In the SN, changes of CD68 contents did not reach statistical significance (F (3,12) = 1.652, *p* = 0.230) (Figure 10c).

## 4. Discussion

In the present study, we used de novo MPTP female monkeys with an extensive DA denervation to model a later stage of PD. Signs of inflammation and glial activity, measured using the markers of reactive astrocytes (GFAP) and microglia (Iba1 and CD68), were observed within several nucleus of the basal ganglia in MPTP monkeys (results summarized in Table 2). In general, exacerbated inflammation was observed in monkeys displaying LID compared to MPTP monkeys. By contrast, MPEP prevented the increased levels of inflammation and glial activity markers compared to that of monkeys without adjunct treatment.

Levels of the Iba1 and GFAP proteins were also measured in the M1 and in the PFCd, representing two output structures of the basal ganglia, to assess whether the inflammatory activity in these MPTP monkeys can extend beyond the intrinsic network of the basal ganglia. Although some evidence in rodent models of PD and in PD patients indicate that the cortical area M1 is involved in the development of LID [54,55], no effect of MPTP lesion was observed on Iba1 or GFAP levels in the M1 and the PFCd, suggesting that the inflammatory activity was probably restricted within the brain structures composing the basal ganglia. Moreover, no change of Iba1 and GFAP levels were measured in the nucleus accumbens suggesting that inflammation was concentrated in brain regions associated with the control of movement. 

### 4.1. Inflammation in PD and LID

Clinical and preclinical studies clearly demonstrated the presence of chronic inflammation in the brain in PD patients [56]. However, the implication and associated mechanisms of this inflammation in the initiation and progression of PD as well as in LID are still not yet well identified. Although inflammatory processes are necessary and probably useful in the early stages of the disease, a sustained inflammatory response for a long period of time can become very harmful to surrounding neurons. The main regulators of inflammation in the brain are astrocytes and microglia; their functions were shown to be altered and detrimental for the survival of DA neurons even before the first signs of clinical symptoms of PD [51,52,57,58,59,60,61,62,63,64]. Emerging evidence suggests that astrocytes alone or with microglia are involved in the inflammatory mechanisms observed at a later stage of PD [65]. The appearance of neurotoxic astrocytes induced by reactive microglia was found in the SN of PD patients [66]. The lost of normal functions of these neurotoxic astrocytes could induce neuronal death suggesting a significant astrocyte-microglia crosstalk in the mechanisms associated with inflammation in PD [66,67]. Astrocytes and microglia interact very closely with each other and their individual but also concerted implications in chronic inflammation remain to be determined [68].

Although we observed increased levels of GFAP associated with astrogliosis in several regions of the basal ganglia nuclei following MPTP administration, our results do not allow to conclude whether these astrocytes have phenotypes that are useful or detrimental for the survival of DA neurons [69]. Moreover, it seems that the increased astrocyte reactivity observed in the present study is not induced by reactive microglia, which are considered to upregulate Iba1 as they adopt a proinflammatory status, since the levels of Iba1 remained unchanged in MPTP monkeys compared to that of control animals in all the brain regions analyzed. Nevertheless, we cannot exclude the possibility that a significant increase in microglial reactivity or transformation toward proinflammatory phenotypes occurred during the MPTP-induced degeneration phase of our experiment and that microglia-astrocyte interactions have lasted until the animals were euthanized.

### 4.2. The Astrocytes Marker GFAP Levels in the Brain of PD and of Animal Models of PD

Results from postmortem astrocytes studies in PD are variable and no clear consensus has emerged on the involvement of astrocytes in the regulation of inflammation. At an early stage of DA degeneration, several preclinical studies showed that astrocyte activity is part of a compensatory mechanisms aimed at rescuing injured DA neurons and ultimately halting the progression of degeneration by preserving remaining neurons [70]. 

Increased GFAP levels were measured in the pre- and post-commissural levels of caudate nucleus and putamen, as well as in the SN and STN of MPTP + L-Dopa dyskinetic monkeys compared to that of intact monkeys. Accordingly, an extensive striatal astrogliosis as well as astrocytic cellular hypertrophy and upregulation of inflammatory markers was demonstrated in the DA-depleted striatum of 6-OHDA-lesioned rats displaying LID compared to that of the unlesioned striatum [71,72]. Interestingly, no sign of astrogliosis was observed in the DA-depleted striatum of animals that received continuous administration of L-Dopa that did not develop dyskinesias [72]. Moreover, chronic L-Dopa administration in 6-OHDA-lesioned rats induced significant increases in nitrite levels as well as immunoreactivity for 3-nitrotyrosine and inducible nitric oxide synthase (iNOS) in the striatum and SNc [71,72,73]. Indeed, nNOS and iNOS inhibitors were shown to attenuate LID in 6-OHDA-lesioned rats, revealing a clear association between the development of LID and the expression of NOS [71,74,75]. In fact, these studies suggest that high levels of neurotoxic NO produced by increased activity of iNOS in reactive astrocytes following chronic L-Dopa treatment could contribute at least in part to the onset and the severity of LID. Dyskinesia scores did not significantly correlate with the levels of GFAP in any brain region investigated, suggesting that in our animal model reactive astrocytes are not directly, but maybe indirectly, linked to microglial mechanisms associated with inflammation and LID. Indeed, the cellular and molecular mechanisms underlying the possible contribution of astrocytes in the onset and development of dyskinesias remain to be elucidated.

### 4.3. The Astrocytes Marker GFAP Levels in the Brain of MPTP Monkeys, LID, and MPEP 

In MPTP monkeys treated with L-Dopa + MPEP inhibiting the development of LID, levels of the astrocytic marker GFAP were significantly increased in the caudate nucleus and putamen as well as in the SN compared to that of controls. These increases tended to be less pronounced than for the MPTP + L-Dopa-treated monkeys, while no significant increase was observed in the STN. mGlu5 receptors are expressed in astrocytes [32], and their activation in primary cultures of rat cortical astrocytes was shown to regulate glutamate transmission by inducing the activation of extracellular glutamate transporter GLT-1, allowing to remove excess extracellular glutamate and thereby preventing excitotoxicity [76]. However, an overexpression and/or sustained activation of mGlu5 receptors was reported to negatively modulate the expression of GLAST (another astrocytic glutamate transporter) and GLT-1 proteins [77]. Interestingly, a study in an astrocytic cell line showed that methamphetamine-induced increased expression levels of the proinflammatory cytokines IL-6 and IL-8 were attenuated following treatment with MPEP [78]. 

### 4.4. Brain Microglia-Mediated Inflammation in PD and MPTP Monkeys 

The present results showed no change of Iba1 levels in MPTP monkeys compared to intact animals in the basal ganglia nuclei analyzed except in the pre-commissural caudate nucleus where a decrease was measured. These results are in agreement, at least in part, with those of the study from Barcia et al. [79] showing that the number of Iba1^+^ microglial cells in the SNc of monkeys rendered parkinsonian with MPTP (male and female) was not changed compared to control animals one to five years after administration of the neurotoxin. However, a detailed microanatomical analysis of microglial cells revealed that microglia are in a reactive phenotype, more precisely in a polarized state oriented toward neighbouring dopaminergic neurons and with persistent phagocytic characteristics [79]. Moreover, no significant change in CD68 levels was observed in MPTP monkeys in the present study compared to that of intact animals in all the basal ganglia nuclei analyzed, which could be interpreted as a normal phagocytic activity of microglia. Also, the decrease or tendency to decrease Iba1 levels in MPTP monkeys, mainly in the caudate and putamen, could probably be explained by the fact that given the high degree of dopaminergic denervation in these animals very few dopaminergic endings remain, requiring less reactive microglia.

In a recent study, we showed in an MPTP monkey model using neurotoxin and L-Dopa treatment regimens similar to the present study that microglial density, measured using Iba1-immunostaining, was increased by 32% compared to that of control monkeys in the putamen at the post-commissural level; whereas, the expression of CD68 in Iba1^+^ cells was similar between MPTP-treated monkeys and control animals [80]. In the present study, no significant change of Iba1 and CD68 levels measured by Western blots were observed in the post-commissural putamen. The monkeys of these two studies had similar parkinsonian scores and survival time post-MPTP. Comparisons with other studies conducted in monkeys showed an increase in the density of reactive microglia in the SNc, but not in the striatum, one year after the last dose of MPTP was administered in cynomolgus monkeys compared to intact monkeys, suggesting that microglial transformation towards a phagocytic state in the surroundings of degenerating dopaminergic neurons persists long after MPTP intoxication [79]. Another study using male macaque monkeys (*Macaca nemistrina*) showed after four to five weeks of chronic low-dose MPTP administration an intense heterogeneous microglial response with phagocytic characteristics as demonstrated by phenotypic changes such as microglia with a swollen perinuclear cytoplasm and fewer and enlarged processes compared to ramified microglia, the presence of numerous multicellular microglia and contiguous fat granules with ramified microglia in the SN, nigrostriatal tract, GPi, and GPe, without evidence of noticeable morphological changes of microglia in the striatum despite a significant loss of TH^+^-neurons [81]. Interestingly, the degree of microglial response does not seem to depend on the level of dopaminergic denervation in monkeys rendered parkinsonian with MPTP [81]. These results clearly show that the dynamics of microglial cell recruitment following injury to nigrostriatal dopaminergic neurons are complex and differ between the various nuclei of the basal ganglia network as a function of time after injury.

The presence of microgliosis in the striatum and SN is also observed in different neurotoxin-based rodent models of PD such as MPTP, 6-OHDA, lipopolysaccharide (LPS), and rotenone as well as in transgenic and protein-based models of PD implicating α-synuclein (reviewed in [56,82]). In a MPTP mouse model of PD, the polarization of microglia towards dopaminergic neurons associated with phagocytic activity was shown to be a transient phenomenon that stops as soon as dopaminergic neurons die [83]. Care should be taken when comparing results of the present and other studies conducted in monkeys and in neurotoxin-based rodent models and examining changes in microglia after administration of different neurotoxins because of several contributing factors such as the dose and administration regimen of the neurotoxin used [84], survival time postlesion, as well as the age and sex of the animals, which can influence microglial function [85,86]. 

Studies in postmortem brain samples of PD patients also revealed a sustained pro-inflammatory environment with an increased density of Iba1^+^ cells measured in the SN. Microglial phenotypes associated with their priming (exacerbated inflammatory and phagocytic response to challenges) and reactivity were also observed in the SN and the putamen. They were associated with increased numbers of MHC class II- and CD68-positive cells, and an increased expression of the intracellular adhesion molecule ICAM-1, integrin receptors CD11a, and scavenger receptor TLR2 [50,87,88,89,90]. Moreover, elevated levels of proinflammatory cytokines and enzymes were measured in the striatum and SN of patients with PD [88,91,92,93,94,95]. These results suggest that inflammation in PD, partly induced by microglia among the SNc and striatum and other nuclei of the basal ganglia, is chronic and was sustained probably for several years (where it could exacerbate the degenerative process) ([96,97] and see reviews: [98,99]). Although the results in humans with PD seem to point in the same direction as those seen in MPTP monkeys, it is important to note that postmortem studies are performed on brains of patients several years after clinical diagnosis of the disease, and that these patients received L-Dopa treatment with probably several adjuvant treatments, in addition to differing in their presence or absence of dyskinesias among other motor fluctuations. Moreover, the degeneration of dopaminergic neurons observed in idiopathic PD is progressive and spreads over several years, while in our monkey model the degeneration is severe and induced rapidly over a period of several months. Therefore, these results cannot be directly compared to the *de novo* MPTP alone monkeys used in our studies. It is instead more appropriate and relevant to compare the studies in humans with our group of MPTP + L-Dopa dyskinetic monkeys. 

### 4.5. Brain Microglia-Mediated Inflammation: Positive Correlation with LID

In dyskinetic monkeys, we found that L-Dopa treatment increased Iba1 levels in the pre-commissural putamen, GPi, GPe, and SN, when compared to intact and MPTP animals whereas the levels of CD68 were increased significantly in the GPi and in the pre-commissural putamen. There are no studies in the literature on the link between inflammation and the development of dyskinesias in the basal ganglia of monkeys except our previous study [80]. It was reported in this study that the increased density of Iba1^+^ microglial cells, cell body, and arborization areas observed in the post-commissural putamen of MPTP monkeys were normalized in dyskinetic monkeys treated with L-Dopa; whereas microglial immunoreactivity for CD68 was unchanged in MPTP monkeys and decreased in those treated with L-Dopa compared to MPTP animals [80]. Differences in survival time after MPTP and methodological approaches could account for the different results obtained in the MPTP + L-Dopa monkeys between these studies. Several preclinical studies in rodent models of PD have shown that treatment with L-Dopa induces or exacerbates inflammation via a direct involvement of microglia releasing pro-inflammatory cytokines promoting an initiation and/or development of dyskinesias [72,100,101,102,103,104,105,106,107,108,109]. In fact, the strong positive correlations between Iba1 levels and dyskinetic scores measured in several basal ganglion nuclei in the present study strongly support a direct link between LID and microglial reactivity. However, the mechanisms linking microglia-mediated inflammation to dyskinesias are not yet elucidated. 

The increase in microglial markers of inflammation in the GPe (Iba1) and GPi (Iba1 and CD68) in dyskinetic monkeys compared to that of vehicle-treated MPTP monkeys, as well as the strong positive correlations between these markers and dyskinetic scores, suggest that the appearance of LID is directly associated with the presence of an inflammatory environment. This is unexpected since there is no evidence of neuronal degeneration in these nuclei in PD [110,111]. Although microglia phagocytoses cellular debris from degenerating neurons, they also have the ability to clear synapses with impaired activities [112]. In fact, maladaptive neuronal plasticity in the striatum following the administration of L-Dopa induces faulty information processing which impacts on the activity of downstream basal ganglia nuclei including the GPe and GPi, where abnormal gamma-aminobutyric acid (GABA)-ergic activity was demonstrated in dyskinetic animals [113]. Hence, it is likely that L-Dopa treatment could alter the activity of GABAergic neurons in the GPe and GPi together with the activity of microglia in these regions.

### 4.6. MPEP Reduced Dyskinesias and Microglial Inflammation Markers in MPTP Monkeys

In MPTP monkeys treated with L-Dopa + MPEP that developed less pronounced LID, the levels of Iba1 were restored to control values in the putamen (pre-commissural level). Putaminal [^3^H]ABP688 specific binding to mGlu5 receptors in these monkeys was found to be elevated in L-Dopa-treated MPTP monkeys compared to control monkeys, but not in those treated with L-Dopa + MPEP [17]. Moreover, mGlu5 receptor levels were increased in the putamen of parkinsonian patients displaying motor complications (LID or wearing-off) compared to those without motor complications [19]. In addition, the dyskinesia scores of the monkeys used in the present study strongly correlated positively with their [^3^H]ABP688 specific binding in the putamen [23] and with the levels of Iba1 in the pre-commissural caudate and putamen, SN, GPe, and GPi as well as with the levels of CD68 in the pre-commissural putamen, suggesting that in this animal model, mGlu5 receptors modulation in the basal ganglia is intimately linked to cellular and molecular mechanisms associated with microglia-mediated inflammation and LID.

Pharmacological blockade of mGlu5 receptors was reported to reduce DA neurons degeneration in various animal models of PD [26,27,44,114,115,116,117,118]. mGlu5 receptors were found in microglia but their outcomes on microglial phenotype and inflammation are not yet well known and controversial [31]. Indeed, several in vitro and in vivo studies showed that activation of microglial mGlu5 receptors induces a decrease in inflammation and production of associated proinflammatory substances in different microglial cell culture and neuropathological models [32,119]. By contrast, chronic treatment of 6-OHDA-lesioned rats with MPEP was shown to have no effect on neuronal survival or on the enhanced microglial CD11b expression and morphological changes as well as on the astroglial increased expression of GFAP protein and morphological changes in the DA-depleted striatum and SNc [120]. Nevertheless, subchronic treatment of rats with MPEP was shown to abolish the MPTP-induced accumulation of OX-6-positive cells in the SNc (indicative of microglial reactivity) and dopaminergic degeneration of the nigrostriatal system [44]. The results of the present study do not allow to determine whether the anti-inflammatory effects of MPEP observed in our MPTP model are induced via an interaction with its neuronal receptor rather than a direct interaction with the microglial mGlu5 receptors. LID are mostly associated with increased glutamate transmission leading to maladaptive synaptic plasticity in the basal ganglia [8]. An up-regulation of putaminal mGlu5 receptors was observed in the MPTP monkeys with LID used in the present study [23] and stimulation of mGlu5 receptors was shown to facilitate glutamate release in the striatum [121]. Therefore, it is likely that the anti-inflammatory effects of MPEP observed in MPTP monkeys with LID are induced via an interaction with its neuronal receptor that are abundantly expressed among the basal ganglia structures. However, we cannot exclude the possibility that the reduction in Iba1 and CD68 levels could be the result of a direct interaction of MPEP with microglial mGlu5 receptors. Indeed, it is possible that in our model the modulation of microglia activity via mGlu5 receptors is different from what is reported in other animal and cellular models of PD, as well as brain regions outside the basal ganglia nuclei in other neuropathologies. However, the mechanisms by which MPEP reduces dyskinesias and inflammation in our animal model is likely multifactorial since we previously showed that the antidyskinetic effect of MPEP was also associated with a normalisation of various markers of both direct and indirect output pathways in the striatum [17,22,23,24,25,122] which could contribute to normalizing inflammation in the different nuclei of the basal ganglia.

### 4.7. General Considerations

Strengths of the present experiment include the use of monkeys since their brain is closer to the human brain compared to rodents. In primates the striatum is separated into the caudate nucleus and putamen whereas in the rodent it is one structure. Moreover, the globus pallidus shows morphological and anatomical differences between rodents and primates with a separation in primates into the GPi and GPe, whereas in rodents they are classified respectively as entopeduncular nucleus and globus pallidus [110]. In addition, the present experiment used de novo MPTP monkeys without prior dopaminergic drug treatments, with a similar parkinsonian score and subsequent treatment with the same dose of L-Dopa. Moreover, the time between MPTP lesioning and euthanasia was the same for all the MPTP monkeys included in the experiment. This allowed to reduce some of the variability of human PD brain studies differing in the duration of the disease, disease stages, drug treatments, and various comorbidities that could affect inflammation markers. The number of monkeys per group in this experiment allowed to clearly show motor behavior differences due to the lesion and treatments, as well as LID between the experimental groups and numerous associated biochemical changes [17,22,23,24,25,122]. However, for the present measurements of inflammation and glial activity markers in some brain regions, additional animals per groups would be required to obtain sufficient statistical power. Table 2 summarizes the percent change *versus* values for control monkeys in the brain regions investigated.

## 5. Conclusions

The present study investigated levels of microglia (Iba1 and CD68) and astrocytes (GFAP) markers of inflammation and gliosis in the brain of MPTP monkeys with different levels of dyskinesias induced by L-Dopa and prevented with an added treatment with MPEP a NAM of mGlu5 receptor. Brain regional differences of GFAP, Iba1 and CD68 were observed concentrated in the basal ganglia without effects in the nucleus accumbens, M1 and PFCd (summarized in Table 2). The putamen, which is implicated in the control of movement, was the brain region displaying highest changes of GFAP in the dyskinetic monkeys and significant positive correlations of Iba1 and CD68 with LID. Elevated GFAP levels but to a lesser extent than in the putamen were observed in the caudate nucleus, SN, and STN of dyskinetic L-Dopa-treated monkeys with no significant increase in the L-Dopa + MPEP group. Positive correlations were also observed between Iba1 levels and LID in the caudate nucleus, globus pallidus and SN but not with CD68. Additional studies are required to investigate microglia-astrocyte interactions in the course of PD progression and development of LID. Treatments acting on microglia (e.g., minocycline, gene colony-stimulating factor 1 receptor (CSF1R) inhibitors) could be used to determine their implication in LID.

## Figures and Tables

**Figure 1 cells-11-00691-f001:**
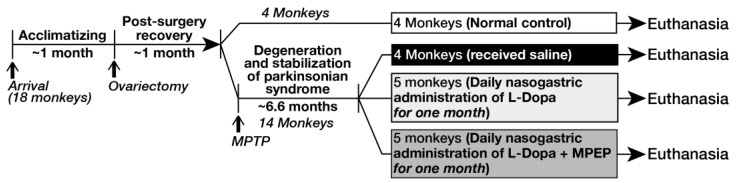
Timelines for animal lesion and treatments.

**Figure 2 cells-11-00691-f002:**
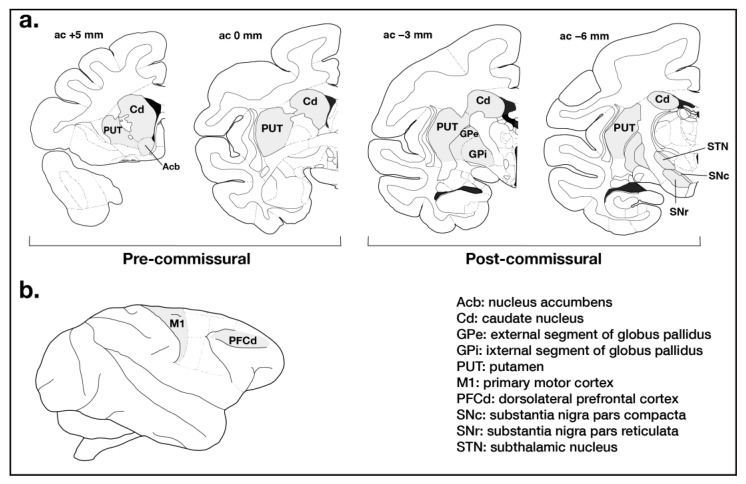
(**a**) Schematic representation of macaque (*macaca fascicularis*) brain showing stereotaxic coordinates [53], where caudate nucleus (Cd), putamen (PUT), nucleus accumbens (Acb), external (GPe) and internal (GPi) segments of globus pallidus, whole substantia nigra (SNc and SNr parts), and subthalamic nucleus (STN) were dissected for Western blot experiments. (**b**) Lateral view of macaque brain showing anatomical regions where primary motor cortex (M1, including regions controlling foot, trunk, and hand) and dorsolateral prefrontal cortex (PFCd) were dissected for Western blot experiments.

**Figure 3 cells-11-00691-f003:**
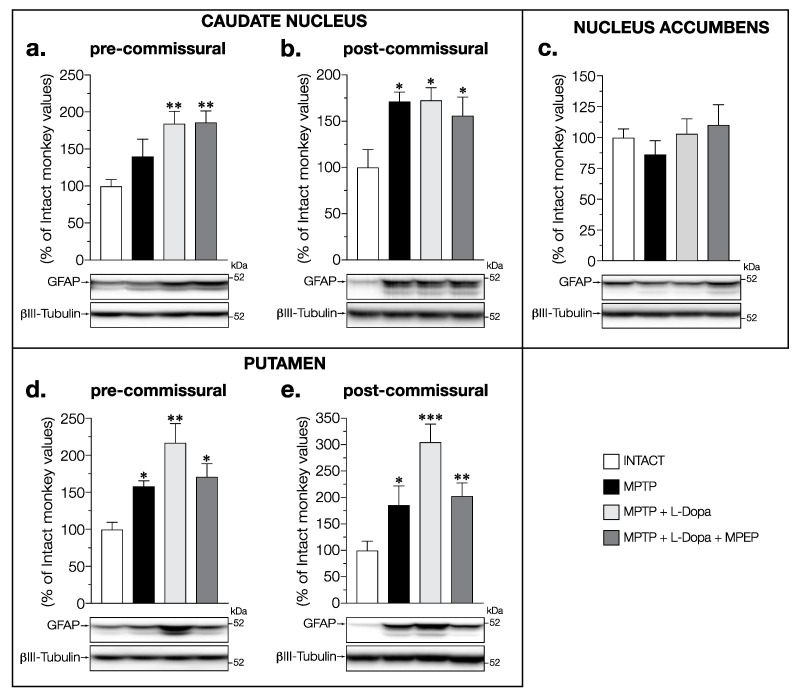
Western blots for GFAP protein levels in (**a**,**b**) caudate nucleus, (**c**) nucleus accumbens, and (**d**,**e**) putamen of intact monkeys (control), MPTP monkeys treated with saline, L-Dopa, and L-Dopa + MPEP. Values are means of arbitrary units expressed as percentage of control (βIII-tubulin was used as internal loading control) of 3–4 independent experiments ± standard error of mean of 4–5 monkeys per group. * *p* < 0.05, ** *p* < 0.01 and *** *p* < 0.001 vs. INTACT.

**Figure 4 cells-11-00691-f004:**
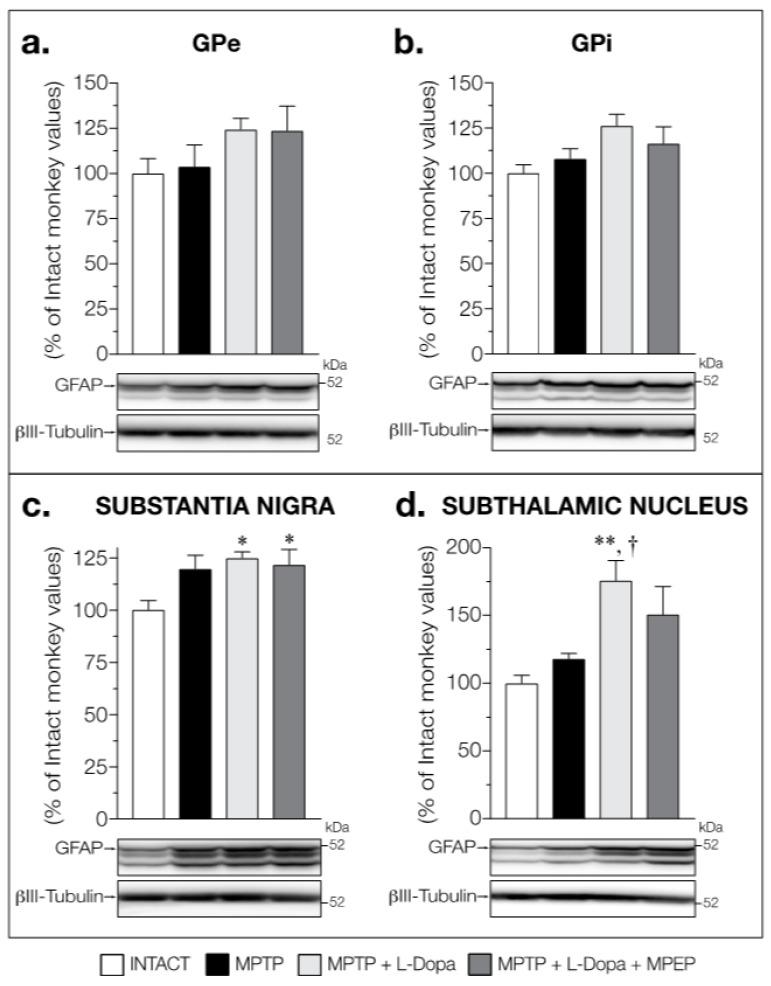
Western blots for GFAP protein levels in (**a**) GPe, (**b**) GPi, (**c**) substantia nigra, and (**d**) subthalamic nucleus of intact monkeys (control), MPTP monkeys treated with saline, L-Dopa, and L-Dopa + MPEP. Values are means of arbitrary units expressed as percentage of control (βIII-tubulin was used as internal loading control) of 2–4 independent experiments ± standard error of mean of 4–5 monkeys per group. * *p* < 0.05 and ** *p* < 0.01 vs. INTACT; † *p* < 0.05 vs. MPTP.

**Figure 5 cells-11-00691-f005:**
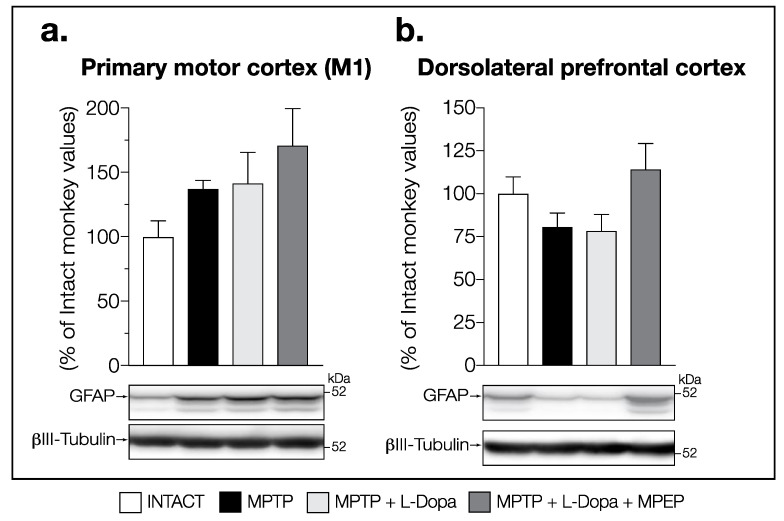
Western blots for GFAP protein levels in (**a**) primary motor cortex (M1) and (**b**) dorsolateral prefrontal cortex (PFCd) of intact monkeys (control), MPTP monkeys treated with saline, L-Dopa, and L-Dopa + MPEP. Values are means of arbitrary units expressed as percentage of control (βIII-tubulin was used as internal loading control) of 4–6 independent experiments ± standard error of mean of 4–5 monkeys per group.

**Figure 6 cells-11-00691-f006:**
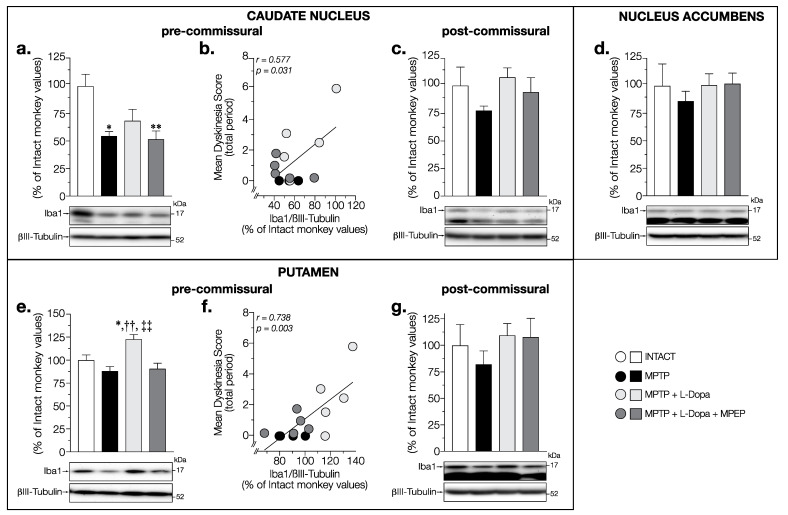
Western blots for Iba1 protein levels in (**a**,**c**) pre- and post-commissural caudate nucleus, (**d**) nucleus accumbens, and (**e**,**g**) pre- and post-commissural putamen of intact monkeys (control), MPTP monkeys treated with saline, L-Dopa, and L-Dopa + MPEP, and correlation between mean dyskinesia score of the MPTP monkeys treated with saline, L-Dopa, and L-Dopa + MPEP, and Iba1 levels in pre-commissural (**b**) caudate nucleus, and (**f**) putamen expressed as percentage of control (βIII-tubulin was used as internal loading control); each point represents an individual monkey. Values are means of arbitrary units expressed as percentage of control (βIII-tubulin was used as internal loading control) of 3–4 independent experiments ± standard error of mean of 4–5 monkeys per group. * *p* < 0.05 and ** *p* < 0.01 vs. INTACT; †† *p* < 0.01 vs. MPTP; ‡‡ *p* < 0.01 vs. MPTP + L-Dopa + MPEP.

**Figure 7 cells-11-00691-f007:**
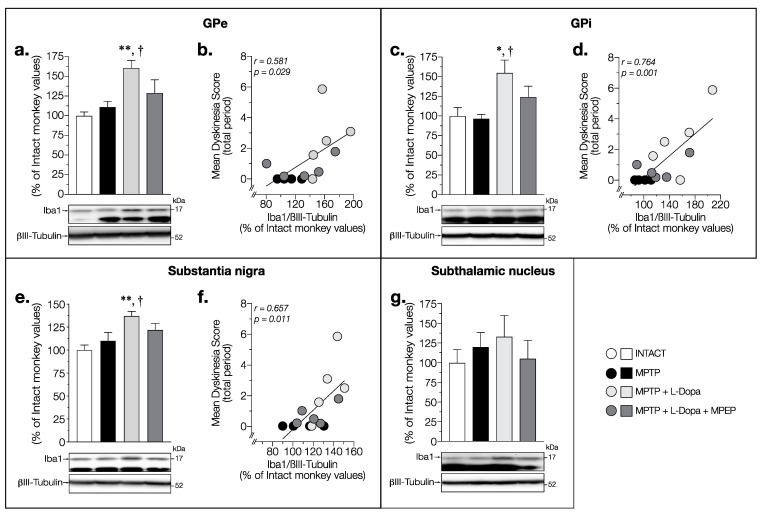
Western blots for Iba1 protein levels in (**a**) GPe, (**c**) GPi, (**e**) substantia nigra, and (**g**) subthalamic nucleus of intact monkeys (control), MPTP monkeys treated with saline, L-Dopa, and L-Dopa + MPEP, and correlation between mean dyskinesia score of MPTP monkeys treated with saline, L-Dopa, and L-Dopa + MPEP, and Iba1 levels in the (**b**) GPe, (**d**) GPi, and (**f**) substantia nigra expressed as percentage of control (βIII-tubulin was used as internal loading control); each point represents an individual monkey. Values are means of arbitrary units expressed as percentage of control (βIII-tubulin was used as internal loading control) of 2–4 independent experiments ± standard error of mean of 4–5 monkeys per group. * *p* < 0.05 and ** *p* < 0.01 vs. INTACT; † *p* < 0.05 vs. MPTP.

**Figure 8 cells-11-00691-f008:**
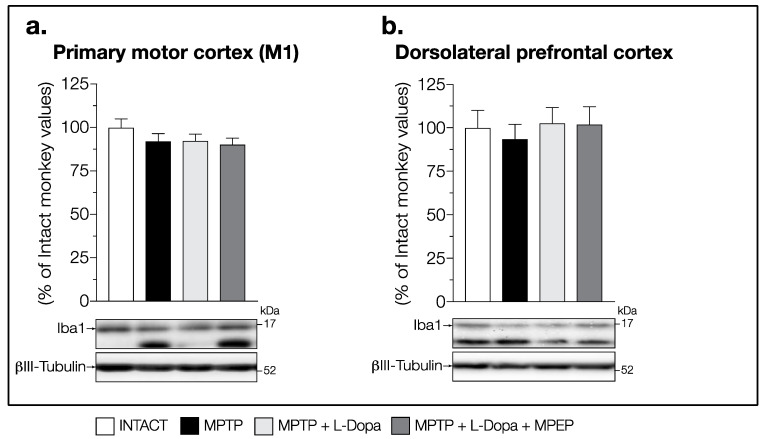
Western blots for Iba1 protein levels in (**a**) primary motor cortex (M1) and (**b**) dorsolateral prefrontal cortex (PFCd) of intact monkeys (control), MPTP monkeys treated with saline, L-Dopa, and L-Dopa + MPEP. Values are means of arbitrary units expressed as percentage of control (βIII-tubulin was used as internal loading control) of 4–5 independent experiments ± standard error of mean of 4–5 monkeys per group.

**Figure 9 cells-11-00691-f009:**
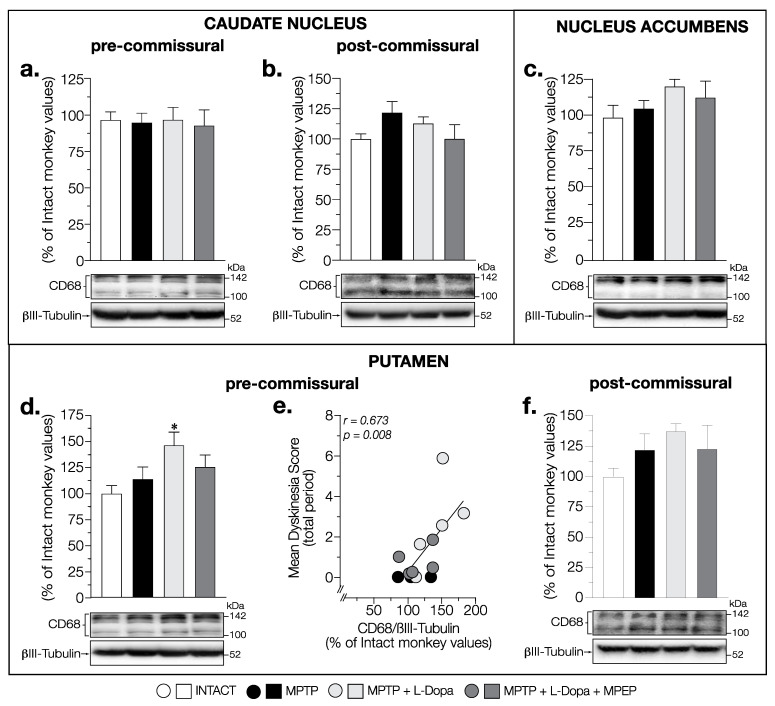
Western blots for CD68 protein levels in (**a**,**b**) the pre- and post-commissural caudate nucleus, (**c**) nucleus accumbens, and in (**d**,**f**) the pre- and post-commissural putamen, of intact monkeys (control), MPTP monkeys treated with saline, L-Dopa, and L-Dopa + MPEP, and correlation between mean dyskinesia score of MPTP monkeys treated with saline, L-Dopa, and L-Dopa + MPEP, and CD68 levels in (**e**) pre-commissural putamen expressed as percentage of control (βIII-tubulin was used as internal loading control); each point represents an individual monkey. Values are means of arbitrary units expressed as percentage of control (βIII-tubulin was used as internal loading control) of 2–4 independent experiments ± standard error of mean of 4–5 monkeys per group. * *p* < 0.05 vs. INTACT.

**Figure 10 cells-11-00691-f010:**
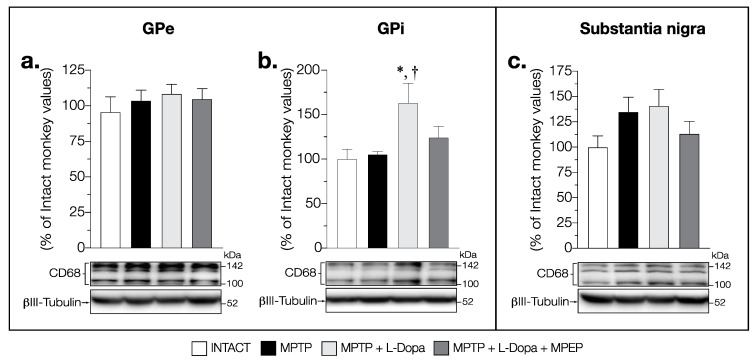
Western blots for CD68 protein levels in (**a**) GPe, (**b**) GPi and (**c**) substantia nigra of intact monkeys (control), MPTP monkeys treated with saline, L-Dopa and L-Dopa + MPEP. Values are means of arbitrary units expressed as percentage of control (βIII-tubulin was used as internal loading control) of 3–5 independent experiments ± standard error of mean of 4–5 monkeys per group. * *p* < 0.05 vs. INTACT; † *p* < 0.05 vs. MPTP.

**Table 1 cells-11-00691-t001:** Description of monkeys included in present experiment.

	INTACT (Control)	MPTP	MPTP + L-Dopa	MPTP + L-Dopa + MPEP
**Animal data**				
Number of monkeys	4	4	5	5
Age (year)	6.2 ± 0.6	6.1 ± 0.2	7.1 ± 0.3	6.0 ± 0.4
Weight (kg)	3.5 ± 0.1	3.6 ± 0.2	3.7 ± 0.3	3.6 ± 0.1
Parkinsonian score ^a^	-	9.5 ± 1.2	10.3 ± 0.9	9.8 ± 0.8
Mean dyskinesias scores (total period)	-	-	3.3 ± 0.9	0.7 ± 0.3 †
Survival time post MPTP (months)	-	3.5 ± 0.9	6.6 ± 1.1	6.2 ± 1.1
**Biochemical data**				
** *Caudate nucleus post-commissural* **				
DA content (ng/mg protein)	112.06 ± 7.12	0.35 ± 0.08 ****	0.96 ± 0.22 ****	0.93 ± 0.30 ****
5-HT content (ng/mg protein)	14.43 ± 0.31	9.86 ± 1.44	10.43 ± 1.3	13.48 ± 1.75
** *Putamen post-commissural* **				
DA content (ng/mg protein)	94.27 ± 3.94	0.79 ± 0.18 ****	2.13 ± 0.52 ****	1.33 ± 0.42 ****
5-HT content (ng/mg protein)	7.70 ± 0.49	7.21 ± 1.10	6.87 ± 0.75	6.91 ± 0.48

^a^ Basal parkinsonian score after administration of vehicle to monkeys the day before the onset of L-Dopa treatment. DA, dopamine; 5-HT, serotonin. **** *p* < 0.0001 vs. control monkeys and † *p* < 0.001 vs. MPTP + L-Dopa-treated monkeys.

**Table 2 cells-11-00691-t002:** Comparative results for the astrocytes marker GFAP and microglial markers Iba1 and CD68 levels in the brain of the experimental groups of MPTP monkeys compared to controls and their relationship with L-Dopa-induced dyskinesias (LID) and their prevention with MPEP.

Brain Region	Treatment														
	**MPTP**				**MPTP + L-Dopa**				**MPTP + L-Dopa + MPEP**				**Correlation LID vs**		
	**GFAP**	**Iba1**	**CD68**		**GFAP**	**Iba1**	**CD68**		**GFAP**	**Iba1**	**CD68**		**GFAP**	**Iba1**	**CD68**
**Caudate nucleus** Pre-commissural Post-commissural	*+40* **+71**	**−45** *−22*	*−1* *+22*		**+84** **+73**	*−31* *+8*	*+1* *+13*		**+86** **+56**	**−48** *−5*	*−3* *0*		no no	positive no	no no

**Putamen** Pre-commissural Post-commissural	**+58** **+86**	*−12* *−21*	*+14* *+22*		**+117** **+205**	**+23** *+5*	**+49** *+37*		**+71** **+105**	*−9* *+3*	*+26* *+23*		no no	positive no	positive no

**Globus pallidus** GPe GPi	*+9* *+8*	*+11* *−3*	*+8* *+5*		*+24* *+26*	**+61** **+55**	*+13* **+57**		*+24* *+12*	*+29* *+24*	*+10* *+22*		no no	positive positive	no no

**Substantia nigra**	*+20*	*+10*	*+35*		**+25**	**+37**	*+41*		**+22**	*+22*	*+13*		no	positive	no

**Subthalamic nucleus**	*+18*	*+20*	ND		**+76**	*+33*	ND		*+45*	*+5*	ND		no	no	ND

% of decreased or increased levels *vs.* respective control monkeys are shown. Data for measures with a significant ANOVA are presented in bold for *p* ≤ 0.05 in post-hoc tests and when the ANOVA was not significant values are shown in italic; LID: L-Dopa-induced dyskinesias (mean dyskinesia scores); ND: Not done due to limited amount of tissue; no significant change was observed in the nucleus accumbens, the primary motor cortex (M1) and the dorsolateral prefrontal cortex (PFCd) for GFAP, Iba1 or CD68 levels and was not included in the table.

## Data Availability

The data presented in this study are available on request from the corresponding author.

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
