# Peer review of "Prevention of L-Dopa-Induced Dyskinesias by MPEP Blockade of Metabotropic Glutamate Receptor 5 Is Associated with Reduced Inflammation in the Brain of Parkinsonian Monkeys"

_cells, 2022, doi:10.3390/cells11040691_

Round 1

Reviewer 1 Report

The present manuscript (ISSN 2073-4409) is an interesting translational study on the role of inflammation in Parkinson’s disease. It strengthens previous preclinical findings showing how non-neuronal inflammatory processes contribute to the pathophysiology of L-DOPA-induced dyskinesia and how the mGLUR5 receptor may provide a potential therapeutic target for LIDs management, by its established anti-inflammatory effects on astrocyte and microglia.

Specifically, Morissette et al., investigated the role of inflammatory responses, in terms of levels of microglia (Iba1 and CD68) and astrocyte (GFAP) markers of inflammation and gliosis, in several basal ganglia nuclei of MPTP-lesioned cynomolgus female monkeys w/ and w/o LIDs. Moreover, they tested the antidyskinetic properties of the negative allosteric modulator of the metabotropic glutamate receptor MPEP, in relation to its modulation on inflammatory pathways affected by MPTP and L-DOPA administration. Authors found increased levels of inflammatory markers primarily in brain regions involved in the control of movement, such as putamen, substantia nigra and globus pallidus, but not in nucleus accumbens (as negative area). They also found a robust positive correlation between dyskinesia scores and microglial markers in these regions. Most importantly, they confirm previous findings from the same group that MPEP significantly blunted dyskinesias scores in MPTP-L-DOPA treated monkeys, and this rescue was paralleled by normalization of some of the inflammatory markers in a region-specific manner.

In my opinion, while partially novel (the authors already published the antidyskinetic effects of MPEP in both rodents and monkeys), this work adds interesting findings for the field and contributes to elucidating the neurobiological underpinnings of the protective effects of MPEP on dyskinesia. As the authors highlighted in the introduction, amantadine is the only drug with sufficient antidyskinetic effect, but its antidyskinetic properties are accompanied by severe side effects, including hallucinations.

Given the elevated expression of mGluR5 in the striatum and other basal ganglia nuclei, and its specific modulatory control of glutamatergic transmission through the basal ganglia circuit, MPEP may provide a very interesting pharmacological tool for the treatment of LIDs. Thus, elucidating the mechanisms underlying its antidyskinetic effects is of great relevance to improve therapeutic applicability.

I have just a few minor comments that may improve the manuscript.

- Based on what I mentioned above, please change the title to a “Research article-manner”. In this form, the title of this manuscript appears to me like a review article. Please also highlight in the title the rescue properties of MPEP exerted by inflammatory mechanisms. In my opinion, this is an important point.  

- Accordingly, please change the structure of the abstract, highlighting the therapeutic potential of mGLUR5 modulation and your previous work on it. Again, the message of the manuscript should be the effects of MPEP on inflammation.

- Please shortened the discussion. For example, there are some paragraphs of the discussion that may belong to the introduction section.

- Please remove some sentences like “a subject of debate to this day”…(raw 429)

- To help the reader, please add to the manuscript a schematic representation of the experimental design and MPEP treatment protocols.

- The sentence… “The detailed behavioral results of this experiment were previously reported (raw 165), what does that mean?

- Please change the title of the subparagraph of the results section in order to describe the outcomes (not the methods) of the experiments. For example Effects of MPTP on GFAP levels of…; MPEP attenuates the increase in GFAP levels induced by L-DOPA in MPTP-lesioned monkeys.

Reviewer 2 Report

The manuscript entitled „ Modulation of inflammation in the brain of parkinsonian monkey with L-Dopa-induced dyskinesias” contains studies of inflammatory markers of astrocytes and microglia in parkinsonian monkeys treated with L-Dopa. In addition, the Authors investigated the neuroprotective and anti-dyskinetic effects of the mGlu5 receptor antagonist, MPEP in MPTP model of Parkinson's disease. The research is very interesting and the results obtained are very promising. The Authors chose the methodology correctly and presented their results in a very clear manner, both in the form of figures and tables. The manuscript is very extensive. The authors measured changes in 3 inflammatory proteins (GFAP, Iba1 and CD68) in many structures of the monkey brain. The discussion chapter is very extensive, and the authors comment on the results obtained in a very detailed manner.

I have some  comments:

  1. Why was the norepinephrine level not measured? Noradrenergic neurons, in addition to dopamine ones, degenerate in Parkinson's disease. The HPLC methodology with electrochemical detection allows the measurement of all monoamines (DA, NA and 5-HT) in one sample. Please comment.
  2. Table 1 shows that animals that received MPTP + L-Dopa or MPTP + L-Dopa + MPEP had very low dopamine levels (similar to the MPTP alone group). L-Dopa is a precursor of dopamine, so after its administration, we should observe a significant increase in dopamine in individual brain structures. These results show that L-Dopa is not working or has not crossed the blood-brain barrier. On the other hand, the Authors observed dyskinesias in L-Dopa treated animals. How can these opposing effects be explained? Please add a comment to the discussion section.
